# A deep learning backcasting approach to the electrolyte, metabolite, and acid-base parameters that predict risk in ICU patients

**Albion Dervishi**◯*

Department of Anesthesiology and Intensive Care Medicine, Medius Clinic Nürtingen, Academic Teaching Hospital of the University of Tübingen, Tübingen, Germany

* albiondervishi@gmail.com

## Abstract

### Background

A powerful risk model allows clinicians, at the bedside, to ensure the early identification of and decision-making for patients showing signs of developing physiological instability during treatment. The aim of this study was to enhance the identification of patients at risk for deterioration through an accurate model using electrolyte, metabolite, and acid-base parameters near the end of patients' intensive care unit (ICU) stays.

### Methods

This retrospective study included 5157 adult patients during the last 72 hours of their ICU stays. The patients from the MIMIC-III database who had serum lactate, pH, bicarbonate, potassium, calcium, glucose, chloride, and sodium values available, along with the times at which those data were recorded, were selected. Survivor data from the last 24 hours before discharge and four sets of nonsurvivor data from 48–72, 24–48, 8–24, and 0–8 hours before death were analyzed. Deep learning (DL), random forest (RF) and generalized linear model (GLM) analyses were applied for model construction and compared in terms of performance according to the area under the receiver operating characteristic curve (AUC). A DL backcasting approach was used to assess predictors of death vs. discharge up to 72 hours in advance.

### Results

The DL, RF and GLM models achieved the highest performance for nonsurvivors 0–8 hours before death versus survivors compared with nonsurvivors 8–24, 24–48 and 48–72 hours before death versus survivors. The DL assessment outperformed the RF and GLM assessments and achieved discrimination, with an AUC of 0.982, specificity of 0.947, and sensitivity of 0.935. The DL backcasting approach achieved discrimination with an AUC of 0.898 compared with the DL native model of nonsurvivors from 8–24 hours before death versus survivors with an AUC of 0.894. The DL backcasting approach achieved discrimination with an AUC of 0.871 compared with the DL native model of nonsurvivors from 48–72 hours before death versus survivors with an AUC of 0.846.

**Data Availability Statement:** Data are available from the https://mimic.physionet.org/.

**Funding:** The author(s) received no specific funding for this work.

**Competing interests:** The authors have declared that no competing interests exist.

## Conclusions

The DL backcasting approach could be used to simultaneously monitor changes in the electrolyte, metabolite, and acid-base parameters of patients who develop physiological instability during ICU treatment and predict the risk of death over a period of hours to days.

## 1. Introduction

To improve patient outcomes, there is an urgent need to promptly recognize patients at risk for deterioration by identifying early laboratory trends and then making decisions in a timely manner [1, 2].

Electrolytes, metabolic parameters (e.g., glucose and lactate) and acid-base balance are crucial in understanding the mechanism of death for critical care patients who deteriorate over a period of hours to days.

The imbalance of these laboratory parameters not only signals physiological instability but can lead to cellular and organ injuries; therefore, they are important markers of cellular and organ function [3, 4].

Therefore, the ultimate purpose of the current study was to evaluate the trends and pathophysiological changes in electrolyte, metabolite, and acid-base parameters leading to death, which would enable us to design a model that could support the clinical decision-making process.

Most critical care patients develop physiological instability, and the initiation of goal-directed therapies to maintain electrolyte, metabolite, and pH homeostasis is critical for such patients' outcomes. Goal-directed therapies can be implemented by setting criteria for the desired state of physiological stability through the creation of a desirable future and then working backward to plan the achievement of this state. To facilitate the development of goal-directed therapies, we developed a backcasting model that implements a retrograde prediction of the risk of death over a period of hours to days and calculates the risk probability based solely on laboratory tests.

If a patient's physiological status deteriorates rapidly, clinical decisions are urgently needed; under these circumstances, point-of-care blood analyzers, which are capable of measuring all parameters in this model, can provide rapid on-site results that can be acted upon immediately [5].

The presence of critical disorders of electrolyte, metabolite, and acid-base parameters that indicate severe underlying pathophysiology was found in patients' admission notes or near the end of the hospital stay for nonsurvivors.

There is sufficient evidence that admission laboratory values are significant indicators of a patient's prognosis and candidate therapies [6–10]. Moreover, a model based solely on laboratory tests (the Laboratory Decision Tree Early Warning Score, LDT-EWS) has been developed to predict patient outcomes after intensive care unit (ICU) admission and in-hospital mortality. The LDT-EWS is based on hemoglobin, urea, creatinine, sodium, potassium, and albumin [7, 8].

Alternatively, there is less evidence of laboratory values as indicators of a patient's prognosis at near the end of a patient's hospital stay [11]. Risk assessment and stratification for physiological deterioration often use different models that are based on various laboratory tests, and most of them predict the risk within 24 hours [11–13].

This subject has been investigated by numerous studies, which have noted a wide range of sensitivities and specificities [4, 5, 9], and few studies have focused on the role of abnormal electrolyte and acid-base parameters in predicting patients' physiological deterioration [14].

We hypothesized that a machine learning backcasting approach based on the database-recorded electrolyte, metabolite, and acid-base parameters of patients near the end of their ICU stays could be used to build a model for risk assessment. However, to date, an evaluation of laboratory values of patients near the end of their ICU stays for these parameters in combination with a risk adjustment model has not been performed.

## 2. Method

This study describes a retrospective investigation of critical care patients obtained from the freely available Medical Information Mart for Intensive Care III (MIMIC-III) database provided by PhysioNet (https://www.physionet.org/). MIMIC-III contains information on 38597 distinct adult patients with 49785 hospital admissions at the Beth Israel Deaconess Medical Center (BIDMC) in Boston, Massachusetts, from 2001 to 2012. The MIMIC-III entries contain monitoring data, records, laboratory test results, procedures, orders, mortality outcomes, and demographics [15].

The electrolyte, metabolite, and acid-base parameters were selected in our models because i) they are commonly available in the ICU from most modern blood analyzers, ii) they are important determinants of physiological stability and outcome, and iii) they are measured for most patients. Specifically, our model used lactate, pH, bicarbonate, potassium, calcium, glucose, chloride and sodium. The algorithms used to develop the model architecture and analysis procedure are presented in Fig 2.

We included all available adults, defined as patients aged 15 to 89 years at the time of ICU admission. The selected laboratory values and their measurement times were extracted for surviving and nonsurviving ICU patients to form two benchmark data subsets. The survivor data subset included the model features from 2821 patients within 24 hours before ICU discharge. The nonsurvivor data subset was extracted from four sets of data showing the model features of ICU patients at 48–72, 24–48, 8–24, and 0–8 hours before death.

The nonsurvivor dataset from 0–8 hours before death consisted of the last values of electrolyte, metabolite, and acid-base parameters of 545 patients, recorded in the MIMIC-III database; this dataset was combined with the survivor data from 0–8 hours form the LEMA 0–8 dataset. The nonsurvivor dataset from 8–24 hours before death included 712 patients and was combined with the survivor data subset to form the EMA 8–24 dataset. The nonsurvivor dataset from 24–48 hours before death included 653 patients and was combined with the survivor data subset to form the EMA 24–48 dataset. The nonsurvivor dataset from 48–72 hours before death contained 426 patients and was combined with the survivor data subset to form the EMA 48–72 dataset.

The distribution of the datasets is expressed as the maximum, minimum and mean ± standard deviation (SD), and the significance of the differences between surviving and nonsurviving patients was assessed using the t test. A bivariate (Pearson's) correlation test and r-package "qgraph" functions were used to produce a visual graphic network of the correlation and interrelationships between parameters [16]. Age and sex were included in the data selection but were not incorporated in the model for risk adjustment.

### Model development

We chose a generalized linear model (GLM), as well as more specific logistic regression-binomial family, random forest (RF) and deep learning (DL) models, which are the most frequently used models for binary classification in medicine. Models were implemented by using the open-source H2O R package. The functions "h2o.deeplearning", "h2o.randomForest" and

"h2o.glm" perform grid searches in succession to obtain the best models with optimal hyper-parameters. For binomial classification problems, logloss was used as the optimization metric.

The model includes 8 parameters, defined as: $X_i$ = {LACTATE$_i$ + PH$_i$ + BICARBONATE$_i$ + POTASSIUM$_i$ + CALCIUM$_i$ + GLUCOSE$_i$ + CHLORIDE$_i$ + SODIUM$_i$}. For the binary response categorical variable $y$, patients who did not survive are coded as 1, and those who did survive are coded as 0, where $y_i \in \{0,1\}$.

The estimation of the GLM logistic regression (binomial) model is $\beta_k$, where $\beta_1 \ldots \beta_8$ is the parameter vector, and $\beta_0$ is the intercept.

The GLM binomial model fitting to an output category can be written as follows, where $\hat{p}$ is the expected probability that the outcome is obtainable:

$$\hat{p} = \frac{e^{X_i^T \beta_k}}{1 + e^{X_i^T \beta_k}}$$

where $X_i^T \beta_k = \beta_0 + \beta_1 \text{LACTATE}_i + \beta_2 \text{PH}_i + \beta_3 \text{BICARBONATE}_i + \beta_4 \text{POTASSIUM}_i + \beta_5 \text{CALCIUM}_i + \beta_6 \text{GLUCOSE}_i + \beta_7 \text{CHLORIDE}_i + \beta_8 \text{SODIUM}_i$

DL (deep structured learning or hierarchical learning) is a subset of machine learning methods based on artificial neural networks; our models applied the feedforward architecture used by H2O. DL architectures are inspired by artificial intelligence models and perform feature extraction in a hierarchical manner similar to the layered learning process of the primary sensory areas of the neocortex in the human brain. DL models can learn good feature representations from raw data and have exhibited high performance with complex data [17, 18]. In recent years, a DL approach has also been widely implemented in areas of bioinformatics such as prediction and prevention of diseases as well as personalized treatment [19]. Our model uses multilayer neural networks as is shown in Fig 1. For the model selected by optimization metrics, the DL automatic supervised training revealed that the two hidden layers with 200 and 200 nodes with rectified linear activation, Bernoulli distribution and regularization procedures L1 and L2 were zero, producing the best performance in terms of classification with logloss of 0.137.

The nonlinear activation function $f(\alpha)$ is used throughout the network, where the weighted combination $\alpha = \sum_i w_i x_i + b$, with bias b, represents the neuron's activation threshold, and $x_i$ and $w_i$ represent the firing neuron's input values and their weights, respectively. Our DL model used a rectified linear activation function: $f(\alpha) = \max(0, \alpha)$ in range $(\cdot) \in \mathbb{R}_+$. This can be interpreted as follows: for each negative of $\alpha$ input, the function returns 0, but the same values are returned back if the value of $\alpha$ is positive [20].

The RF is widely used in medicine as a machine learning algorithm based on decision-tree theory for solving classification problems. The RF generates a forest of classification trees rather than a single classification, where each tree classifier is generated by using random sample observations from training data and recursively partitioning data based on values of the predictor variables. Our data in the binary response RF utilizes a single tree to calculate the survival probability of patients and then computes the probability of nonsurvival as $1.0 - p0$. The selected random forest model had 50 trees with logloss of 0.306 as optimization metrics.

For testing purposes, the datasets were divided into the training and testing sets (60% and 40%, respectively). The DL, RF and GLM models were trained. Subsequently, the test data were used to analyze the performance of the model. The performance of the models was compared among the LEMA 0–8, EMA 8–24, EMA 24–48, and EMA 48–72 datasets.

Additionally, univariate DL analyses of lactate, pH, bicarbonate, potassium, calcium, glucose, chloride, and sodium were performed to explore the individual importance of each of those variables in risk prediction based on nonsurvivors 0–8 hours from death versus

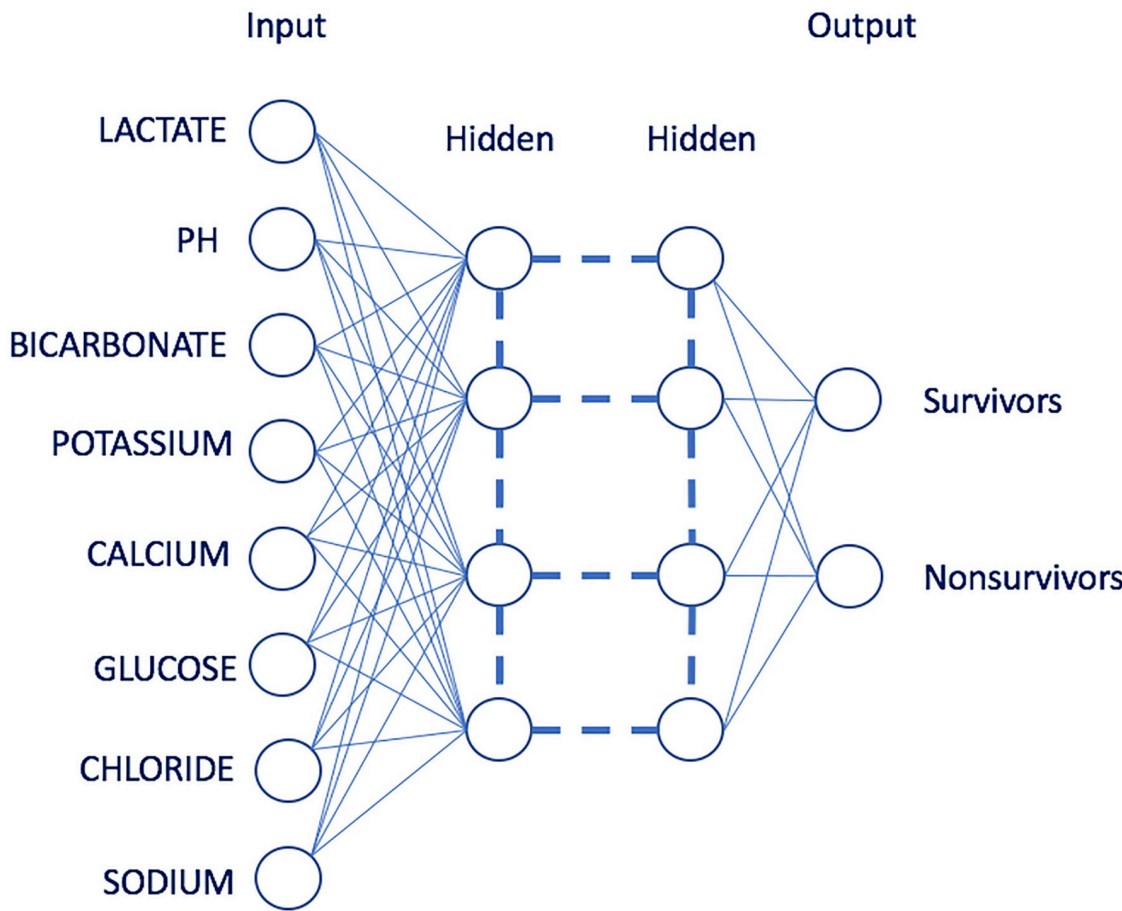

**Fig 1. DL with two hidden layers of neurons.** The input neurons are parameters of LEMA datasets, and output neurons indicate the survival and nonsurvival probability.

survivors. It is reasonable to calculate severity scores from the probability of ICU patient deterioration derived from multivariate and univariate DL analyses.

The receiver operating characteristic (ROC) curve was plotted to define the optimal cutoff value for discrimination and to analyze the ability of univariate and multivariate parameters to predict patient survival or nonsurvival. Moreover, we estimated the area under the ROC curve (AUC) to evaluate the accuracy of the DL, RF, and GLM models. The AUC ranged from 0 to 1, where good discrimination is suggested to correspond with AUCs of 0.8–0.9, and values >0.9 represent very good discrimination performance. The value at the top left, "**closest.topleft**", where the sensitivity and specificity curves intersected, was considered the optimal cutoff value for the ROC curve, defined as $min((1 - sensitivity)^2 + (1 - specificity)^2)$ [21].

## Backcasting approach algorithm

The backcasting approach algorithm of the DL, RF and logistic regression models involves **"working backward"** from the endpoint of discharge or death (LEMA 0–8) up to 72 hours prior. To achieve this goal, we initially trained the native models of the LEMA 0–8, EMA 8–24, EMA 24–48, and EMA 48–72 datasets and then conducted retrospective predictions of the DL LEMA 0–8 model by using test data from the EMA 8–24, EMA 24–48, and EMA 48–72 datasets. We compared the performance of the backcasting models with that of the native models

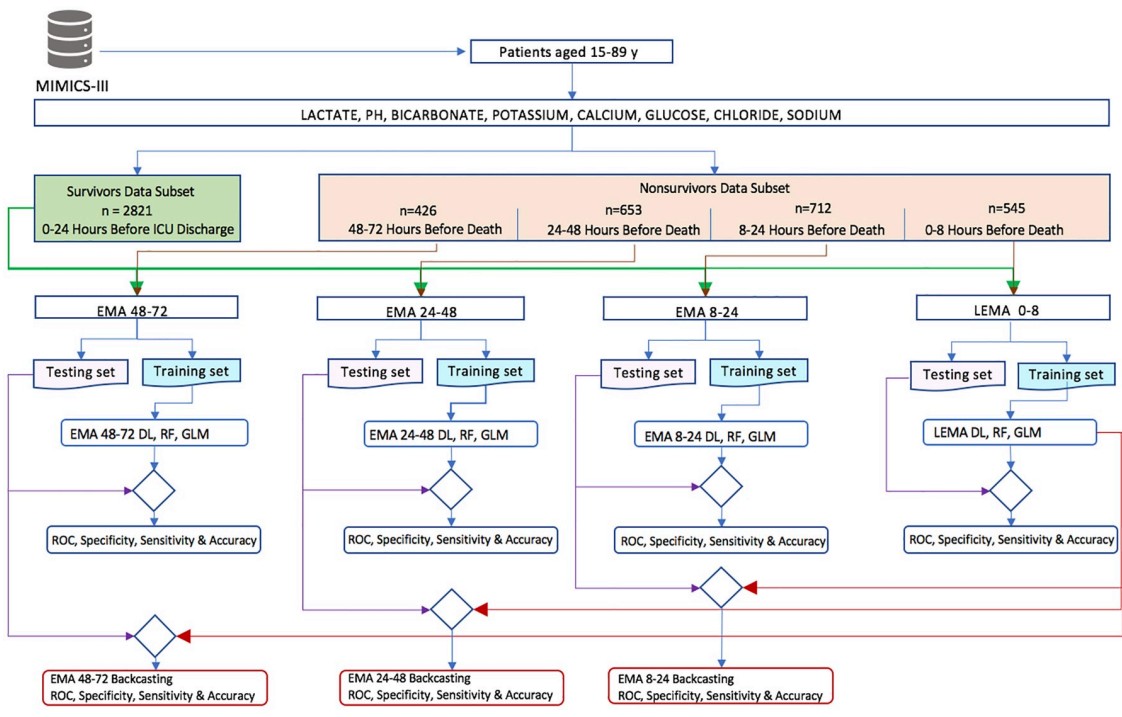

**Fig 2. Algorithm flowchart for model construction.**

during the patients' last 72 hours in the ICU. The algorithm flowchart for model construction and performance is shown in Fig 2 and Table 4.

All analyses were conducted using the statistical software R, and the queries were stored in a public GitHub repository [22].

## 3. Results

A total of 5157 patients were included in the study. The electrolyte, metabolite, and acid-base parameters of the 2821 survivors within 24 hours before ICU discharge were compared with those of nonsurvivors as of 48–72 hours (426 patients), 24–48 hours (653 patients), 8–24 hours (712 patients) and 0–8 hours (545 patients) before the time of death. The values of the patients' electrolyte, metabolite, and acid-base parameters are shown in Tables 1 and 2 as well as Fig 2.

As shown in Tables 1 and 2 and Fig 3, in the nonsurvivors, lactate showed a significant increasing trend, with mean ± SD values of 3.6±3.3, 4.4±3.9, 5.2±4.8 and 8.7±5.7 at 48–72, 24–48, 8–24 and 0–8 hours before the time of death, respectively (P < 0.001). The same trend was true for glucose, with mean ± SD values of 145.9±69.2, 154.1±83.3, 157.2±105.4 and 188.5 ±130.6, respectively, in the same time windows (P < 0.001). Minor increasing trends in sodium and potassium were observed among the nonsurvivors compared with the survivors, with p-values less than 0.05. pH and bicarbonate measurements showed decreasing trends in the nonsurvivors compared with the survivors, with p-values less than 0.001. The mean ± SD values of pH (7.3±0.1 vs.7.2±0.2) and bicarbonate (21.5±5.9 vs. 17.1±7.3 mEq/dL) in nonsurvivors also decreased from the first time window (48–72 hours from death) to the last time window (0–8 hours from death).

We investigated Pearson's (bivariate) correlation between electrolyte, metabolite, and acid-base parameters between ICU patients who remained stable and those who physiologically

**Table 1. Data are presented as the maximum, minimum and mean±SD parameter values and times for survivors 0–24 hours before ICU discharge vs. nonsurvivors 24–48 and 48–72 hours before death.**

| | Survivors | | | Nonsurvivors | | | | Nonsurvivors | | | |
| --- | --- | --- | --- | --- | --- | --- | --- | --- | --- | --- | --- |
| | 0–24 Hours Before ICU Discharge | | | 24–48 Hours Before Death | | | | 48–72 Hours Before Death | | | |
| | Time | Min-Max | Mean± SD | Time | Min-Max | Mean± SD | p-value | Time | Min-Max | Mean± SD | p-value |
| Patients | | | n = 2821 | | | n = 653 | | | | n = 426 | |
| Age (years) | | | 59.6±17.1 | | | 65.5±15.4 | <0.001 | | | 64.5±15.6 | <0.001 |
| SEX (F) | | | n = 1151 | | | n = 283 | | | | n = 172 | |
| (M) | | | n = 1670 | | | n = 370 | | | | n = 254 | |
| LACTATE (mmol/L) | 15.51 h±6.81 h | 0.3–14.4 | 1.5±1.0 | 35.05 h±6.38 h | 0.6–28 | 4.4±3.9 | <0.001 | 59.46 h±6.47 h | 0.5–27 | 3.6±3.3 | <0.001 |
| PH (units) | 12.20 h±6.67 h | 7.03–7.62 | 7.4±0.1 | 36.03 h±5.98 h | 6.73–7.63 | 7.3±0.1 | <0.001 | 60.18 h±6.07 h | 6.82–7.54 | 7.3±0.1 | <0.001 |
| BICARBONATE (mEq/dL) | 10.65 h±6.41 h | 8.0–52 | 25±4.6 | 36.65 h±6.01 h | 5.0–45 | 20.9±5.9 | <0.001 | 60.76 h±5.94 h | 5.0–47 | 21.5±5.9 | <0.001 |
| POTASSIUM (mEq/dL) | 9.91 h±6.82 h | 1.9–7.3 | 4.1±0.5 | 36.64 h±6.04 h | 2.2–9.8 | 4.4±0.9 | <0.001 | 60.78 h±5.96 h | 1.9–8.8 | 4.3±0.8 | <0.001 |
| CALCIUM (mg/dL) | 10.98 h±6.43 h | 5.8–11.9 | 8.3±0.7 | 36.60 h±6.04 h | 4.1–15 | 8.3±1.2 | 0.161 | 60.72 h±6.01 h | 3.8–13.2 | 8.3±1.1 | 0.267 |
| GLUCOSE (mg/dL) | 10.57 h±6.44 h | 19–866 | 131.7±51.7 | 36.67 h±6.03 h | 25–726 | 154.1±83.3 | <0.001 | 60.75 h±5.97 h | 28–560 | 145.9±69.2 | <0.001 |
| CHLORIDE (mEq/L) | 10.48 h±6.49 h | 82–132 | 105.2±5.4 | 36.65 h±6.02 h | 71–129 | 103.2±8 | <0.001 | 60.75 h±5.94 h | 72–135 | 103.9±8 | <0.001 |
| SODIUM (mEq/L) | 10.35 h±6.58 h | 116–159 | 138.9±4 | 36.64 h±6.02 h | 116–161 | 137.9±6.2 | <0.001 | 60.74 h±5.94 h | 120–159 | 138.5±6 | 0.086 |

The p-values show the significance of the differences between surviving and nonsurviving patients.

deteriorated and died within 0–8 hours. Moderate to weak correlation correlations were identified from the correlation matrix (Table 3 and Fig 4). Pearson's correlation (r) measures the linear correlation between two variables X and Y and has the value +1 and −1. The value +1 represents a positive linear correlation and vice versa, and there is no linear correlation when the value is 0.

We found a moderate positive linear correlation between lactate and potassium (r = 0.4), while lactate had a moderate negative correlation with pH (r = -0.6) and bicarbonate (r = -0.5).

Sodium had a moderate positive linear correlation with chloride (r = 0.59), and chloride showed a weak negative correlation with bicarbonate (r = -0.31).

Table 4 shows the performance of the DL, RF and GLM models in the EMA 8–24, EMA 24–48, and EMA 48–72 datasets compared with the predictions of the native models in terms of AUC, sensitivity, specificity, and accuracy. DL outperformed RF and GLM in almost all our models.

EMA 8–24 backcasting, EMA 24–48 backcasting, and EMA 48–72 backcasting using the DL and RF models achieved better AUCs than the native models; however, GLM showed by EMA 24–48 and EMA 48–72 backcasting somewhat lower AUCs than the native models.

Moreover, DL and RF backcasting models had superior discrimination, especially in terms of sensitivity.

The sensitivity of DL-EMA 8–24 backcasting was 0.832, compared with 0.808 for the native model; the sensitivity of DL-EMA 24–48 backcasting was 0.839, compared with 0.767 for the native model; and the sensitivity of DL-EMA 48–72 backcasting was 0.787, compared with 0.757 for the native model.

**Table 2. Data are presented as the maximum, minimum and mean±SD parameter values and times for survivors 0–24 hours before ICU discharge vs. nonsurvivors 8–24 and 0–8 hours before death.**

| | Survivors | | | Nonsurvivors | | | | Nonsurvivors | | | |
|---|---|---|---|---|---|---|---|---|---|---|---|
| | 0–24 Hours Before ICU Discharge | | | 0–8 Hours Before Death | | | | 8–24 Hours Before Death | | | |
| | Time | Min-Max | Mean± SD | Time | Min-Max | Mean± SD | p-value | Time | Min-Max | Mean± SD | p-value |
| Patients | | | n = 2821 | | | n = 545 | | | | n = 712 | |
| Age (years) | | | 59.6±17.1 | | | 64.9±15.9 | <0.001 | | | 66.5±15.1 | <0.001 |
| SEX (F) | | | n = 1151 | | | n = 218 | | | | n = 298 | |
| (M) | | | n = 1670 | | | n = 327 | | | | n = 414 | |
| LACTATE (mmol/L) | 15.51 h±6.81 h | 0.3–14.4 | 1.5±1.0 | 3.49 h±2.27 h | 0.7–27.7 | 8.7±5.7 | <0.001 | 14.65 h±4.45 h | 0.05–26 | 5.2±4.8 | <0.001 |
| PH (units) | 12.20 h±6.67 h | 7.03–7.62 | 7.4±0.1 | 3.45 h±2.30 h | 6.35–7.71 | 7.2±0.2 | <0.001 | 14.81 h±4.31 h | 6.84–7.71 | 7.3±0.1 | <0.001 |
| BICARBONATE (mEq/dL) | 10.65 h±6.41 h | 8.0–52 | 25±4.6 | 4.23 h±2.28 h | 5–65.0 | 17.1±7.3 | <0.001 | 14.98 h±4.46 h | 5.0–46 | 19.6±6.1 | <0.001 |
| POTASSIUM (mEq/dL) | 9.91 h±6.82 h | 1.9–7.3 | 4.1±0.5 | 4.19 h±2.27 h | 1.9–9.6 | 4.8±1.1 | <0.001 | 15.00 h±4.47 h | 2.4–8.5 | 4.5±0.9 | <0.001 |
| CALCIUM (mg/dL) | 10.98 h±6.43 h | 5.8–11.9 | 8.3±0.7 | 4.21 h±2.29 h | 1.8–33.9 | 8.4±2.4 | <0.001 | 14.95 h±4.48 h | 4.2–18.6 | 8.2±1.2 | <0.001 |
| GLUCOSE (mg/dL) | 10.57 h±6.44 h | 19–866 | 131.7±51.7 | 4.23 h±2.28 h | 4–923 | 188.5±130.6 | <0.001 | 15.00 h±4.48 h | 13–1390 | 157.2±105.4 | <0.001 |
| CHLORIDE (mEq/L) | 10.48 h±6.49 h | 82–132 | 105.2±5.4 | 4.22 h±2.28 h | 64–135 | 104.7±9.4 | 0.191 | 14.98 h±4.46 h | 72–137 | 103.5±8.1 | <0.001 |
| SODIUM (mEq/L) | 10.35 h±6.58 h | 116–159 | 138.9±4 | 4.22 h±2.27 h | 112–172 | 140.1±7.7 | 0.004 | 15.00 h±4.45 h | 114–174 | 138.1±6.5 | <0.001 |

The p-values show the significance of the differences between surviving and nonsurviving patients.

To compare the performance of the multivariable DL, RF, and GLM models for probability classification between patients who died within 0–8 hours and those who survived (the LEMA 0–8 dataset), we calculated the optimal cutoff probability value, assessed where the sensitivity and specificity curves intersected and performed AUC analysis.

For DL, the closest top-left cutoff was 0.0659, with an AUC of 0.982, specificity of 0.947 and sensitivity of 0.935. For the random forest, with an AUC of 0.968, specificity of 0.938 and sensitivity of 0.921, the optimal probability cutoff was 0.271. The optimal probability cutoff for the GLM was 0.127, with an AUC of 0.966, specificity of 0.939 and sensitivity of 0.907.

As shown in Table 5, the values of electrolyte, metabolite, and acid-base parameters in the LEMA 0–8 dataset at the model's probability cutoff were found to be similar to the corresponding standard critical values [15]. The DL cutoff threshold values show the closest values to the corresponding standard critical alert values.

The performance of the univariate DL probability classification in the LEMA 0–8 dataset models was evaluated using the AUC, and the "closest.topleft" cutoff was determined from the specificity and sensitivity shown in Table 6.

## 4. Discussion

Electrolytes, metabolites, acids, and bases are continually moving among the intracellular and extracellular compartments of the body, which are in a state of homeostasis maintained by multiple homeostatic mechanisms. The novelty of the proposed backcasting approach is the early identification of failure in the mechanisms that maintain electrolyte, metabolite, and pH homeostasis in ICU patients; monitoring these processes plays a pivotal role in managing the

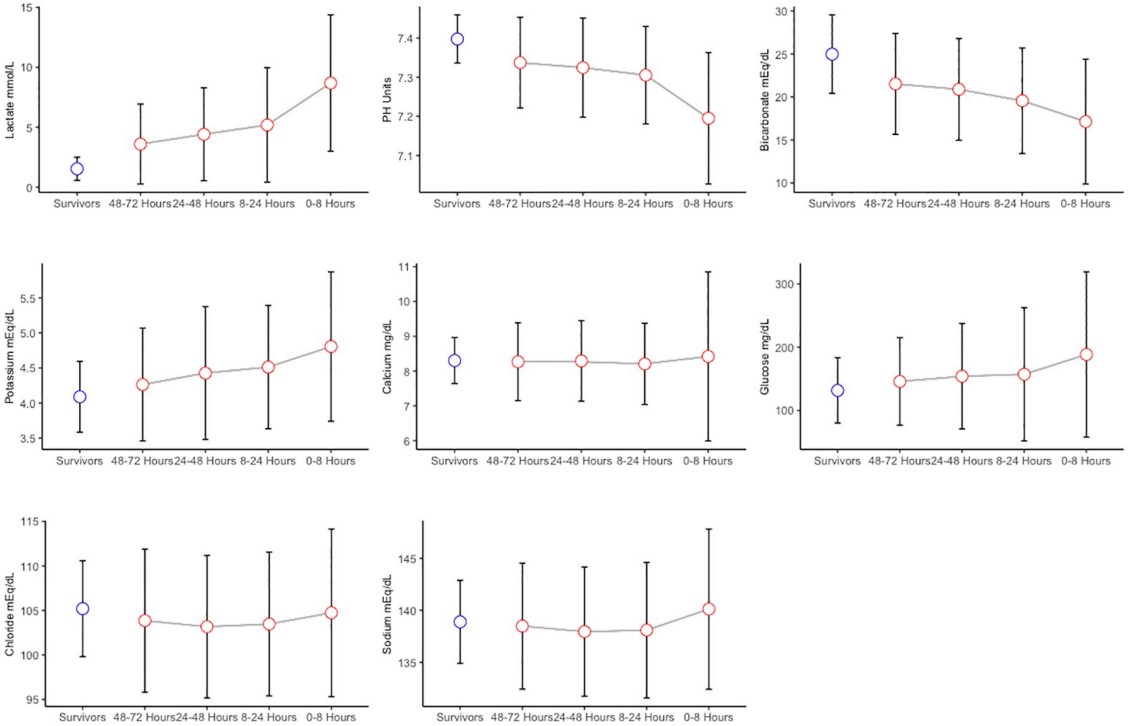

**Fig 3. Mean (standard error) parameter values; blue dots represent the survivors, and red dots represent the nonsurvivors.** The gray line shows the trend of the laboratory test results during the last 72 hours of life in nonsurviving ICU patients.

patient's plan of care. Additionally, early identification of trends in laboratory values can support clinicians in decision-making.

There are many studies in which backcasting methodologies provide a superior approach to complex problems and represent an alternative method for environmental, economic, and time-series analysis, and these methods can be applied in medicine as well [23–25].

The backcasting prediction in our model was developed by performing several investigations. The principal aim was to achieve a risk assessment model with high predictive power. Then, after back-prediction and comparison with native models, we generated the complete backcasting model.

According to Table 5, the performance of the DL, RF and GLM models shows that the LEMA 0–8 AUCs were 0.982, 0.968 and 0.966 and that higher classification and prediction

**Table 3. Bivariate (Pearson's) correlation matrix for electrolyte, metabolite, and acid-base parameters of the LEMA dataset.**

|  | SODIUM | POTASSIUM | CHLORIDE | BICARBONATE | GLUCOSE | CALCIUM | LACTATE | PH |
|---|---|---|---|---|---|---|---|---|
| SODIUM | 1 |  |  |  |  |  |  |  |
| POTASSIUM | -0.11 | 1 |  |  |  |  |  |  |
| CHLORIDE | 0.59 | -0.1 | 1 |  |  |  |  |  |
| BICARBONATE | 0.1 | -0.31 | -0.31 | 1 |  |  |  |  |
| GLUCOSE | -0.05 | 0.11 | -0.17 | -0.14 | 1 |  |  |  |
| CALCIUM | 0.15 | 0.12 | -0.16 | 0.16 | 0.07 | 1 |  |  |
| LACTATE | 0.11 | 0.4 | -0.14 | -0.5 | 0.32 | 0.12 | 1 |  |
| PH | -0.08 | -0.41 | -0.09 | 0.51 | -0.24 | 0.04 | -0.6 | 1 |

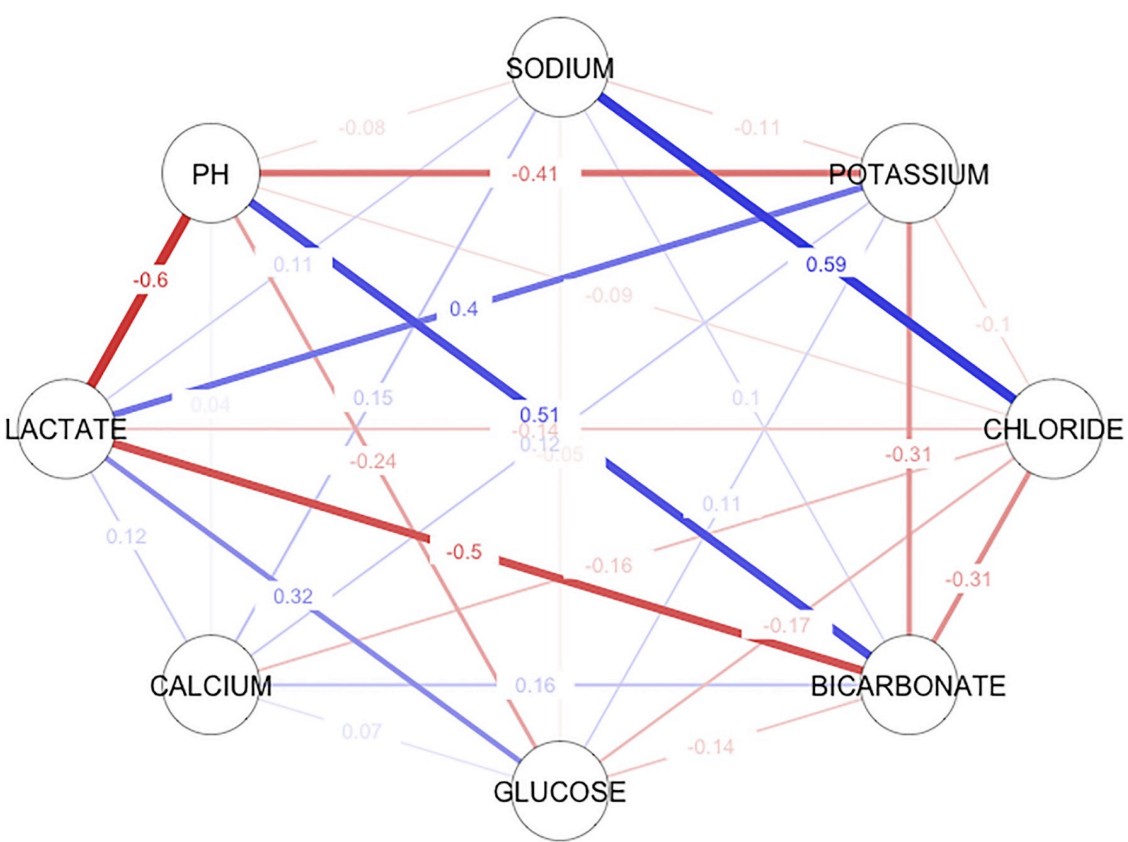

**Fig 4. Correlation network of the bivariate (Pearson's) correlation of the electrolyte, metabolite, and acid-base parameters between ICU patients who remained stable and those who physiologically deteriorated and died in 0–8 hours.** Blue lines represent positive correlations, while red lines represent negative correlations. The thicker and darker a line is, the stronger the correlation.

accuracy were achieved compared with the EMA 8–24 AUCs (0.894, 0.872, and 0.874), EMA 24–48 AUCs (0.873, 0.897 and 0.873) and EMA 48–72 AUCs (0.846, 0.841 and 0.839). Similar results were also derived from a retrospective study through laboratory tests that demonstrated the ability to predict the patient's death within 24–48 hours, with an AUC of 0.88 [11].

The DL and RF backcasting approaches outperformed the native models in estimating risk and distinguishing nonsurvivors from survivors up to 72 hours in advance.

The DL-EMA 8–24 AUC compared with the DL-EMA 8–24 backcasting AUC was 0.894 vs. 0.898; the DL-EMA 24–48 AUC compared with the DL-EMA 24–48 backcasting AUC was 0.873 vs. 0.892; and the DL-EMA 48–72 AUC compared with the DL-EMA 48–72 backcasting AUC was 0.846 vs. 0.871.

GLM preserved the predictive power of the backcasting approach, but its performance was slightly inferior to that of the native models. The GLM-EMA 8–24 AUC compared with the GLM-EMA 8–24 backcasting AUC was 0.874 vs. 0.874; the GLM-EMA 24–48 AUC compared with the GLM-EMA 24–48 backcasting AUC was 0.873 vs. 0.862; and the GLM-EMA 48–72 AUC compared with the GLM-EMA 48–72 backcasting AUC was 0.839 vs. 0.818.

We think the reason why GLM had lower performance than the native models is the lower flexibility of GLM in handling nonlinearity. Alternatively, the excellent performance of the DL backcasting approach can be explained by its high flexibility in nonlinear classification.

**Table 4. Performance of the DL, RF and GLM models and backcasting of the LEMA 0–8, EMA 8–24, EMA 24–48, and EMA 48–72 datasets.**

|  | AUC (95% CI) | Specificity | Sensitivity | Accuracy |
|---|---|---|---|---|
| DL-LEMA 0–8 | 0.982(0.973–0.991) | 0.947 | 0.935 | 0.945 |
| RF-LEMA 0–8 | 0.968(0.953–0.983) | 0.938 | 0.921 | 0.936 |
| GLM-LEMA 0–8 | 0.966(0.95–0.981) | 0.939 | 0.907 | 0.934 |
| DL-EMA 8–24 | 0.894(0.868–0.914) | 0.840 | 0.805 | 0.815 |
| RF-EMA 8–24 | 0.872(0.846–0.898) | 0.852 | 0.781 | 0.838 |
| GLM-EMA 8–24 | 0.874(0.848–0.90) | 0.863 | 0.771 | 0.844 |
| **DL-EMA 8–24 Backcasting** | **0.898(0.875–0.921)** | **0.827** | **0.832** | **0.828** |
| **RF-EMA 8–24 Backcasting** | **0.888(0.862–0.913)** | **0.859** | **0.808** | **0.849** |
| **GLM-EMA 8–24 Backcasting** | **0.874(0.847–0.848)** | **0.856** | **0.771** | **0.838** |
| DL-EMA 24–48 | 0.873(0.840–0.907) | 0.821 | 0.767 | 0.811 |
| RF-EMA 24–48 | 0.897(0.926–0.926) | 0.845 | 0.797 | 0.836 |
| GLM-EMA 24–48 | 0.873(0.838–0.908) | 0.813 | 0.827 | 0.815 |
| **DL-EMA 24–48 Backcasting** | **0.892(0.867–0.928)** | **0.838** | **0.839** | **0.838** |
| **RF-EMA 24–48 Backcasting** | **0.896(0.865–0.928)** | **0.870** | **0.821** | **0.860** |
| **GLM-EMA 24–48 Backcasting** | **0.862(0.825–0.899)** | **0.827** | **0.785** | **0.819** |
| DL-EMA 48–72 | 0.846(0.812–0.902) | 0.797 | 0.757 | 0.792 |
| RF-EMA 48–72 | 0.841(0.807–0.875) | 0.814 | 0.733 | 0.804 |
| GLM-EMA 48–72 | 0.839(0.803–0.875) | 0.758 | 0.787 | 0.762 |
| **DL-EMA 48–72 Backcasting** | **0.871(0.841–0.902)** | **0.820** | **0.787** | **0.816** |
| **RF-EMA 48–72 Backcasting** | **0.850(0.814–0.887)** | **0.849** | **0.751** | **0.836** |
| **GLM-EMA 48–72 Backcasting** | **0.818(0.78–0.857)** | **0.764** | **0.727** | **0.759** |

As shown from our results, the proposed DL and RF backward prediction of the electrolyte, metabolite, and acid-base parameters when used in combination (multivariate analysis) has a better ability to predict the patient's deterioration or death within hours to days.

Moreover, an evaluation of the trends and laboratory values for nonsurvivors during their last 72 hours in the ICU, as shown in Fig 3, showed values of stepwise progression of laboratory abnormalities that led to organ injury up to death. Therefore, predictive backcasting approaches were designed to detect even small changes in the electrolyte, metabolite, and acid-base values, which follow a simultaneous change in multiple parameters, to help guide the clinical decision-making process.

Point-of-care blood analyzers play an important role in risk prediction in this model. These modern blood analyzers are highly capable tools for assessing patients' physiological status,

**Table 5. Parameter threshold values of multivariable DL, RF and GLM models of the LEMA 0–8 dataset at the optimal probability cutoff (closest to the top left) compared with standard critical values.**

|  | Threshold-GLM (Values) | Threshold-RF (Values) | Threshold-DL (Values) | Critical Values |
|---|---|---|---|---|
| LACTATE (mmol/L) | 0.3–7.0 | 0.3–6.7 | 0.3–4.6 | >4 |
| PH (units) | 7.22–7.71 | 7.22–7.59 | 7.19–7.59 | <7.20 and >7.60 |
| BICARBONATE (mEq/dL) | 12–47 | 14–47 | 12–47 | <10 and >40 |
| POTASSIUM (mEq/dL) | 2.3–6.0 | 2.3–6.4 | 2.5–6.4 | < 2.5 and >6.2 |
| CALCIUM (mg/dL) | 6.2–33.1 | 6.2–11.6 | 6.2–11.6 | <6.5 and >13.0 |
| GLUCOSE (mg/dL) | 40–530 | 46–530 | 46–530 | <45 and >450 |
| CHLORIDE (mEq/L) | 82.0–134 | 90.0–129.0 | 82.0–129 | <80 and >120 |
| SODIUM (mEq/L) | 116.0–158 | 126.0–156 | 118.0–150 | <120 and >160 |

**Table 6. Univariate DL analysis in the LEMA 0–8 dataset.**

|  | AUC(95% CI) | Top-Left Threshold (Prob) | Threshold (Values) | Normal Values | Specificity | Sensitivity | Accuracy |
|---|---|---|---|---|---|---|---|
| LACTATE | 0.92(0.889–0.951) | 0.01 | 0.3–2.7 mmol/L | <2 mmol/L | 0.897 | 0.826 | 0.887 |
| PH | 0.867(0.831–0.902) | 0.03 | 7.32–7.52 units | 7.35-7-45 units | 0.899 | 0.764 | 0.878 |
| BICARBONATE | 0.842(0.807–0.877) | 0.07 | 21–33 mEq/L | 24–30 mEq/L | 0.823 | 0.75 | 0.812 |
| POTASSIUM | 0.770(0.731–0.809) | 0.06 | 3.4–4.4 mEq/dL | 3.5–5 mEq/dL | 0.749 | 0.666 | 0.735 |
| CALCIUM | 0.709(0.668–0.750) | 0.03 | 7.9–9 mg/dL | 8–10.5 mg/dL | 0.636 | 0.675 | 0.642 |
| GLUCOSE | 0.683(0.622–0.744) | 0.05 | 85–160 mg/dL | 70–130 mg/dL | 0.737 | 0.606 | 0.717 |
| CHLORIDE | 0.659(0.616–0.702) | 0.1 | 102–111 mEq/L | 95–106 mEq/L | 0.676 | 0.602 | 0.665 |
| SODIUM | 0.619(0.574–0.665) | 0.08 | 137–143 mEq/L | 135–145 mEq/L | 0.631 | 0.549 | 0.619 |

The DL-derived optimal probability cutoff was used to estimate the parameter values discriminating ICU patients who remained stable from those who physiologically deteriorated and died in 0–8 hours. Lactate was the most reliable predictor value, with an AUC of 0.92, followed by pH and bicarbonate, with AUCs of 0.867 and 0.842, respectively.

and they may facilitate the identification of patients at deterioration risk to enable the prompt initiation of goal-directed therapy [26].

A bivariate (Pearson's) correlation between ICU patients who remained stable and those who physiologically deteriorated and died in 0–8 hours is presented in Table 3 and Fig 4, showing simultaneous changes in all the electrolyte, metabolite, and acid-base parameters. This helps us to understand the relationships between variables and provides insight into the complexities of homeostatic mechanisms. The degree of intercorrelations varied from positive to negative among parameters, with pH showing the greatest correlation with the measured lactate r = -0.6, bicarbonate r = 0.51, and potassium r = -0.41, while parameters such as glucose r = -0.26, calcium r = 0.04, chloride r = -0.09 and sodium r = -0.08 revealed the least association. A retrospective, observational study revealed similar results by assessing the correlation between HCO3 and blood pH in pediatric subjects, with r = 0.413 [27].

Multivariate analysis enables risk estimation and statistical assessment of the relationship of electrolyte, metabolite, and acid-base parameters between survivors and nonsurvivors among ICU patients. As shown in Table 5, we found that most of the cutoffs (closest to the top left) for parameter values from DL, RF and GLM models of the LEMA-based multivariate model at a cutoff probability were quite similar to the critical value thresholds. The critical values of laboratory measurements are well established in the medical field for identifying patients who face an imminent or sustained increased risk of death [28, 29]. This approach opens the possibility of defining critical value thresholds by the probability at the upper left of the ROC curve for distinguishing between nonsurvivors and survivors.

Due to the importance of the DL model for the LEMA dataset for the backcasting approach, univariate analyses of the electrolyte, metabolite, and acid-base parameters were performed to examine the contributions of individual variables to risk identification and prediction. All parameters assessed by univariate analysis except lactate displayed similar nonlinear U-shaped relationships with the DL-based risk of death within 8 hours.

## Acid-base parameters

Acid-base disorders are common in ICU patients and are often complex to diagnose and manage appropriately; they are also associated with increased morbidity and mortality [30]. In our model, we focused on the pH value and the metabolic component that reflects the serum bicarbonate level.

Systemic pH homeostasis is maintained between 7.35 and 7.45; this balance is achieved through multiple buffer systems and compensatory mechanisms in which the kidneys and lungs play central roles. Similar to normal pH values, the cutoffs from our univariate analyses are between 7.32–7.52 units. It is well known that pH fluctuations can add to the negative effects of the causative condition, sometimes leading to mortality or complicating cardiopulmonary resuscitation in survivors [30, 31]. A pH of less than 7.2 is recognized as being associated with the initiation of enzyme and protein dysfunctions, including coagulation disorders; this value corresponds to the lower pH threshold value (7.19) of our multivariate model cutoff, as well as established critical values [4]. The pH values in the univariate analysis between survival and mortality within 8 hours achieved a discriminatory AUC of 0.867, with a specificity of 0.89 and sensitivity of 0.76, as shown in Fig 5 and Table 6.

The dynamic balance of bicarbonate is critical for the physiological pH buffering system. Bicarbonate is a byproduct of the body's metabolism; the lungs participate in removing this ion through volatile equilibrium, and the kidneys support homeostasis by active reabsorption and excretion [32]. In ICU patients, bicarbonate is a good predictor of acidosis and mortality [19] as well as acute kidney injury [33]; in our univariate analysis, it showed an AUC of 0.84, with a specificity of 0.82 and a sensitivity of 0.75, as presented in Fig 5 and Table 6.

## Metabolites

Glucose and lactate are the two most essential metabolite parameters used in the monitoring of patients in the ICU [34, 35]. The serum lactate level is established to be an important indicator of

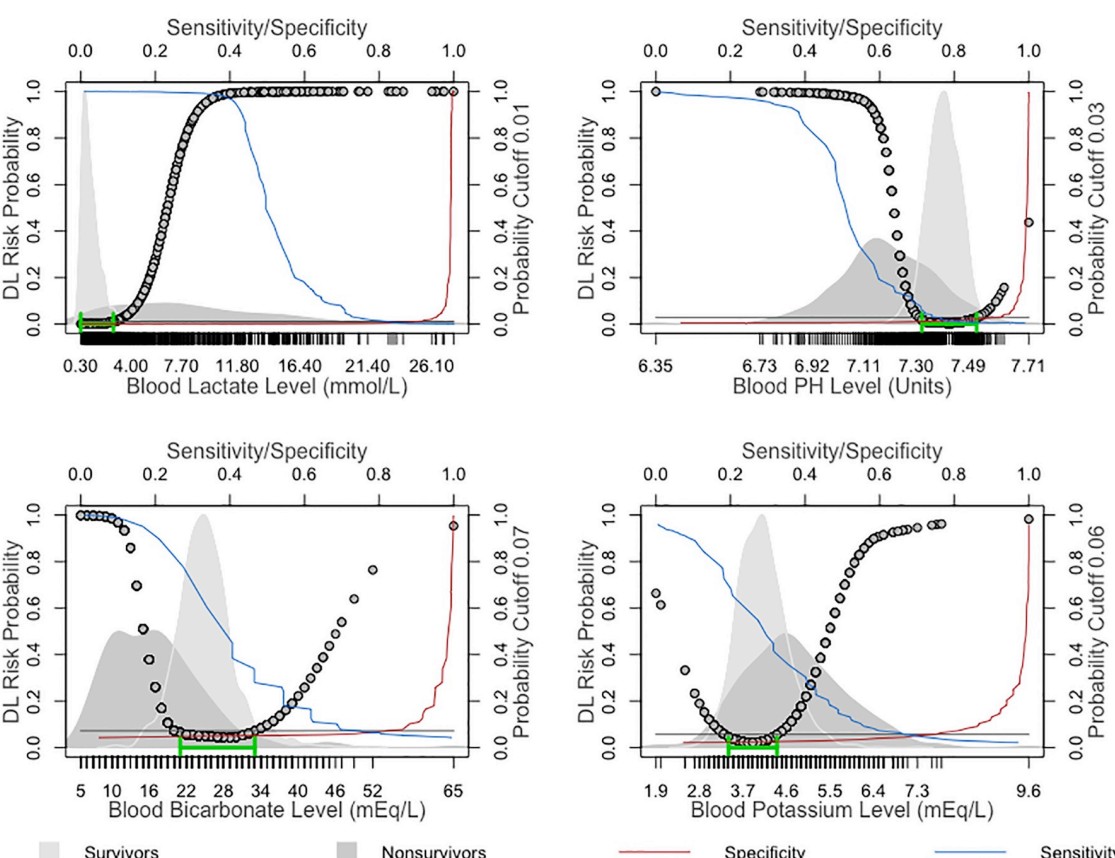

**Fig 5. Univariate analysis depicting the nonlinear association of lactate, pH, bicarbonate, and potassium values with the DL-derived probability of death within 8 hours based on surviving and nonsurviving ICU patients.**

circulatory impairment and, thereby, oxygenation status in critical care [35]. The serum lactate concentration is also a useful predictor for assessing the risk of sepsis, mortality versus survival, and poor versus good neurological outcomes after cardiac arrest [36, 37]. Multiple studies have confirmed that, above 2 mmol/L, the lactate level has a strong positive linear relationship with hospital mortality [36, 38]. Similar results were obtained from the univariate analyses in our study, with a cutoff of 2.7 mmol/L. Additionally, the admission lactate level achieved a pooled AUC of 0.77 as a predictor of cardiac arrest outcomes in a systematic review and meta-analysis [39]. In univariate analyses between survival and death within 8 hours, we found that lactate was an impressive independent predictor, with an AUC of 0.92, a specificity of 0.89, and a sensitivity of 0.82, as shown in Fig 5 and Table 6. Additionally, in multivariate analyses, the optimal lactate cutoff was found to be 4.6 mmol/L, which is close to the established critical value of 4.0 mmol/L [40].

Glucose is used for the diagnosis and surveillance of diabetes mellitus and other metabolic dysfunctions in ICU patients. Glucose variability, such as hyperglycemia and hypoglycemia, is associated with increased mortality and poor outcomes in ICU patients [41, 42]. In our model, risk increased when the glucose level was outside the range of 85–160 mg/dL. However, the optimal cutoff point from multivariate DL in the LEMA dataset was 46.0 mg/dL, which is the critical value alert threshold for hypoglycemia. Univariate analysis with the DL model in the LEMA dataset achieved an AUC of 0.68, with a specificity of 0.73 and sensitivity of 0.60, as shown in Fig 6 and Table 6.

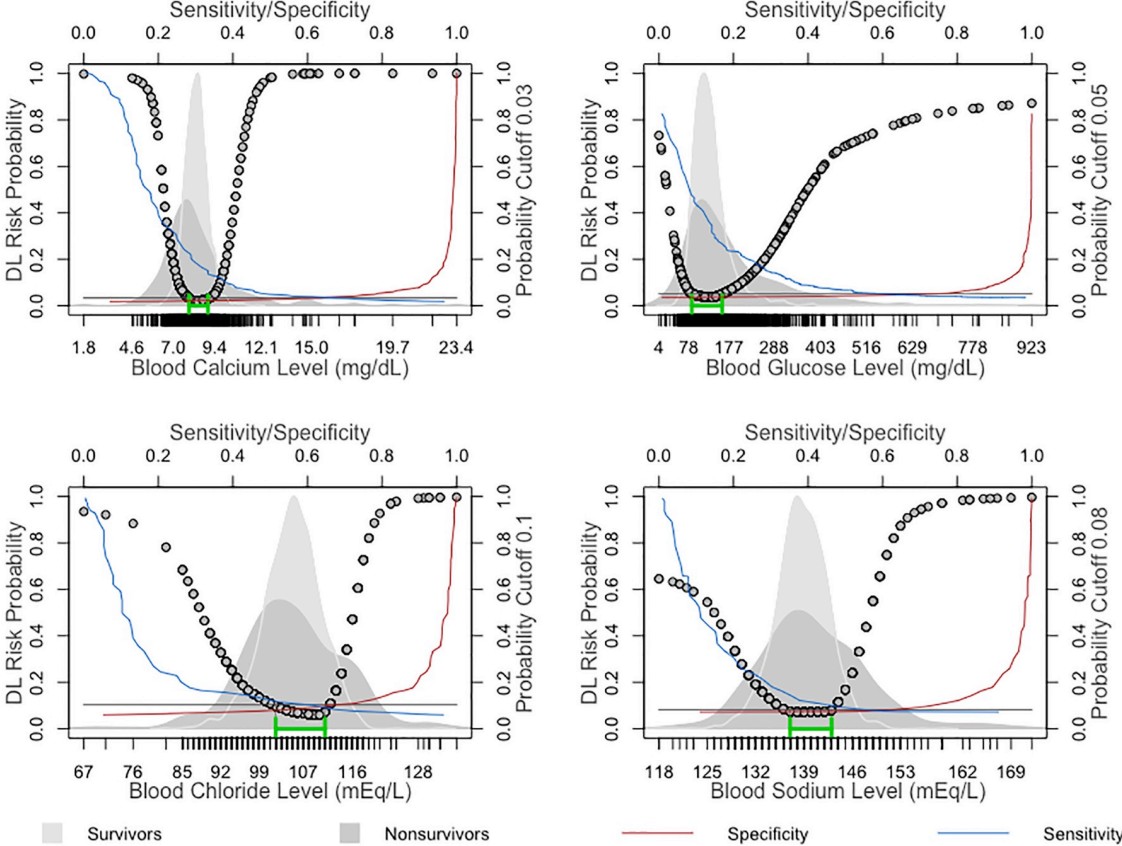

**Fig 6. Univariate analysis depicting the nonlinear association of calcium, glucose, chloride and sodium values with the DL-derived probability of death within 8 hours based on surviving and nonsurviving ICU patients.** The background of each plot represents the density of estimates of the relevant variable among survivors and nonsurvivors. The green line segment indicates the optimal DL-derived range of probability cutoff points below the intersection of the sensitivity and specificity curves, along with the corresponding laboratory values.

The background of each plot represents the density of estimates of the relevant variable among survivors and nonsurvivors. The green line segment indicates the optimal DL-derived range of probability cutoff points below the intersection of the sensitivity and specificity curves, along with the corresponding laboratory values.

## Electrolytes

Disturbances of electrolytes in the ICU patient population have been associated with prolonged ICU stays as well as increased morbidity and mortality [14, 43–45]. Our results suggest that, with even modest departures from normal electrolyte concentrations, risk increases in a nonlinear manner as the magnitude of the disturbance increases. Severe electrolyte disturbances associated with acid-base status, metabolic disorders, or enzyme systems of excitable tissues (including nerves and muscles) are potentially fatal if they cause arrhythmia [44, 46].

Potassium is a major intracellular cation and plays a significant role in action potentials, acid-base homeostasis, and metabolism. The relationship of the potassium concentration and potassium variability with outcomes on admission and in the first 24 hours of the ICU stay is now well established [47, 48]. A critical potassium level (less than 2.5 and more than 6.2 mmol/L) can be life-threatening and requires urgent medical attention; almost identical potassium values (2.5 and 6.4 mmol/L) were obtained as the cutoffs of our multivariate model. Additionally, in our univariate analysis, potassium achieved an AUC of 0.77, specificity of 0.74, and sensitivity of 0.66, as shown in Fig 5 and Table 6.

Sodium is the most significant extracellular cation and plays an important role in serum osmolality and water balance. Disorders of sodium are common in clinical settings, and the importance of sodium as a parameter in the ICU is well recognized [49, 50]. Our results from univariate analyses show cutoffs close to the upper and lower bounds of the normal sodium range: 137–143 mEq/L vs. 135–145 mEq/L. In addition, the critical sodium limits are 120 mEq/L or less and 160 mEq/L or more, corresponding to the multivariate cutoff in our model. In our model, sodium as an individual variable had an AUC of 0.61, specificity of 0.63, and sensitivity of 0.54, as shown in Fig 6 and Table 6.

Chloride is a major extracellular anion and assists in maintaining osmolarity and acid-base balance. Serum chloride alterations in the ICU are usually a result of an underlying condition or secondary to therapeutic interventions [51] and are associated with poor clinical outcomes, increased mortality, and prolonged hospital stays [52]. Our results from univariate analyses show that chloride alterations outside the range of 102–111 mEq/L are related to an increased death risk. Moreover, in the univariate analyses, chloride presented an AUC of 0.65, specificity of 0.67, and sensitivity of 0.60, as shown in Fig 6 and Table 6.

Serum calcium (in its unbound form) is a cation that plays an important role in many physiological processes, such as cell signaling, neurotransmission, muscle contraction, and coagulation. Disturbances in calcium homeostasis are common in ICU settings and are associated with increased mortality in critically ill patients [53, 54]. In our univariate analysis, hypo- and hypercalcemia were associated with an increased risk, with an AUC of 0.70, specificity of 0.63, and sensitivity of 0.67, as displayed in Fig 6 and Table 6.

## Backcasting approach for everyday application

In accordance with the concept of the model backcasting approach for everyday application in ICU facilities, our model can support clinicians in timely decision-making. The four criteria of backcasting according to Holmberg and Larsson are [55]:

1. Identify the criteria that need to be met in a sustainable future.

2. Identify gaps between the current situation and desired situation specified in step 1.

3. Envision the future solution.

4. Identify strategies for achieving sustainability.

The first step of our backcasting approach was defined in the future sustainable desired state of the physiological stability of ICU patients. Our results show that the cutoffs based on multivariate DL models of LEMA 0–8 dataset values, which are quite similar to the standardized critical values, are essential in the discrimination of stable vs. unstable ICU patients. Moreover, DL in the LEMA 0–8 dataset yielded optimal probability cutoffs for univariate parameters that were similar to the normal values.

In step 2, the gap in the recorded electrolyte, metabolite, and acid-base parameters between physiologically unstable and stable ICU patients can accurately be assessed by our backcasting risk model.

With respect to step 3 and step 4, our backcasting model supports the same goal-directed approach to therapy that follows all current medical guidelines to restore electrolyte, metabolite, and acid-base parameters to normal values.

It is crucial for the four points to work together in addressing the simultaneous changes in electrolyte, metabolite, and acid-base parameters to reduce patients' risk.

## 5. Conclusion

We have shown that the backcasting machine learning approach for evaluating electrolyte, metabolite, and acid-base parameters yields better discrimination than day-to-day parameter prediction.

A new method could be used to evaluate and monitor these parameters dynamically, and it also has the potential to identify patients at risk for deterioration such that the medical staff can make decisions and undertake treatments in a timely manner.

## Supporting information

**S1 Appendix.**
(TXT)

## Author Contributions

**Conceptualization:** Albion Dervishi.

**Formal analysis:** Albion Dervishi.

**Funding acquisition:** Albion Dervishi.

**Investigation:** Albion Dervishi.

**Methodology:** Albion Dervishi.

**Validation:** Albion Dervishi.

**Visualization:** Albion Dervishi.

**Writing – original draft:** Albion Dervishi.

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
