## [Decision Letter · Decision Letter 0]

20 Aug 2020

PONE-D-20-09103

A Deep Learning Backcasting Approach to the Electrolyte, Metabolite, and Acid-Base Parameters That Predict Risk in ICU Patients

PLOS ONE

Dear Dr. Dervishi,

Thank you for submitting your manuscript to PLOS ONE. After careful consideration, we feel that it has merit but does not fully meet PLOS ONE’s publication criteria as it currently stands. Therefore, we invite you to submit a revised version of the manuscript that addresses the points raised during the review process.

We look forward to receiving your revised manuscript.

Kind regards,

Feng Luo

Academic Editor

PLOS ONE

Journal Requirements:

2. To comply with PLOS ONE submission guidelines, in your Methods section, please provide additional information regarding your statistical analyses. For more information on PLOS ONE's expectations for statistical reporting, please see https://journals.plos.org/plosone/s/submission-guidelines.#loc-statistical-reporting.

Reviewers' comments:

Reviewer's Responses to Questions

**Comments to the Author**

1. Is the manuscript technically sound, and do the data support the conclusions?

Reviewer #1: Yes

2. Has the statistical analysis been performed appropriately and rigorously? 

Reviewer #1: Yes

3. Have the authors made all data underlying the findings in their manuscript fully available?

Reviewer #1: Yes

4. Is the manuscript presented in an intelligible fashion and written in standard English?

Reviewer #1: Yes

5. Review Comments to the Author

Reviewer #1: The manuscript titled, “A Deep Learning Backcasting ...” studies a deep learning model for prediction of risk using critical care data. The claim is that a ‘bakcasting’-based approach can increase the efficacy of clinical decision making using deep learning. The study is an addition to the already existing knowledge in AI based approaches utilizing clinical data.

The manuscript is lacking in several aspects. I suggest the author revise the manuscript taking into account the comments and suggestions below.

1) The author has been rather frugal in providing explanations and detailing the study. Given the broad readership of the journal, the manuscript needs an overhaul. I would suggest the author to adhere to the standard writing style of Plos one.

2) Related to above, the Abstract needs to be written in a way that clearly states the motivation and introduces the problem, analyses techniques and discusses the main findings. The current format is not suitable.

3) The Introduction should be extended to provide a motivation for the study while discussing the previous research work, what has been lacking therein, and why the current study should be considered as an improvement.

4) Inside the methodology the author needs to elaborate on the technique of ‘backcasting’. Also the flow diagram, Fig. 1 should be discussed more. In general, the backcasting technique needs to be separately motivated. One would also like to understand, in brief, the usual practices of manually processing such data by medical personnel - how these particular numbers are useful in monitoring the health of a patient. I understand the above has been elaborated in the Discussion (referring to Table 3). But would be better to include in the methodology, at least some of it. The author has actually referred to Table 4 & 5 before Table 3.

5) In the Discussion and conclusion the author should again highlight the importance and shortcomings of the study, and be more concrete as to why “deep learning model in recognizing patients at risk for deterioration needs clinical validation”. Are there confounding factors?

6) Correlation aspects between the features have not been discussed. The author should perform more exploratory analysis of the dataset like PCA, etc.

7) “Our model uses multilayer neural networks, but a precise evaluation of their architecture would be beyond the scope of this study” - but a precise description of the model architecture/details and chosen parameters and hyper-parameters is a must.

8) The authors need to compare the deep learning method with some generic baseline model, or some less complex model like Naive Bayes or logistic regression.

9) Why not use something like permutation feature importance?

10) How does the age and gender of the patients (training set) influence the prediction?

6. PLOS authors have the option to publish the peer review history of their article (what does this mean?). If published, this will include your full peer review and any attached files.

Reviewer #1: No

---

## [Author Response · Author response to Decision Letter 0]

14 Oct 2020

Dear Editor, 

I wish to submit a revised manuscript " A Deep Learning Backcasting Approach to the Electrolyte, Metabolite, and Acid-Base Parameters That Predict Risk in ICU Patients.” for PLOS ONE. 

Manuscript Number: PONE-D-20-09103

I am very grateful to you and the reviewer team comments and thoughtful suggestions. 

Based on these comments and suggestions, I made careful revision to the original manuscript. 

Here are the responses to the reviewers’ comments and suggestions.

The reviewer’s comment 1. The author has been rather frugal in providing explanations and detailing the study. Given the broad readership of the journal, the manuscript needs an overhaul. I would suggest the author to adhere to the standard writing style of Plos one.

Response 1. We are very grateful to your comments on the manuscript. By this revision, additional complementary features were written. I wrote additional explanations of the methodology and discussion regarding the model also I tried to follow the standard writing style of Plos one.

The reviewer’s comment 2. Related to above, the Abstract needs to be written in a way that clearly states the motivation and introduces the problem, analyses techniques and discusses the main findings. The current format is not suitable.

Response 2. We thank you very much for your comments and advice. I rewrote the Abstract that states the main findings, results, or arguments of the project.

The reviewer’s comment 3. The Introduction should be extended to provide a motivation for the study while discussing the previous research work, what has been lacking therein, and why the current study should be considered as an improvement.

Response 3. Thank you for this suggestion. In the Introduction, we supplemented facts, references and detection time of critical laboratory values. However, based on our knowledge to date, building a model-based electrolyte, metabolite, and acid-base parameters of patients near the end of their ICU stays has not been performed.

The reviewer’s comment 4. Inside the methodology the author needs to elaborate on the technique of ‘backcasting’. Also, the flow diagram, Fig. 1 should be discussed more. In general, the backcasting technique needs to be separately motivated. One would also like to understand, in brief, the usual practices of manually processing such data by medical personnel - how these particular numbers are useful in monitoring the health of a patient. I understand the above has been elaborated in the Discussion (referring to Table 3). But would be better to include in the methodology, at least some of it. The author has actually referred to Table 4 & 5 before Table 3.

Response 4. Thank you for pointing this out. I tried to address better this issue. The backcasting technique is written separately in the Methods as “Backcasting approach algorithm “. In the Discussion, I discussed more, how I developed the backcasting model. Additionally, in the discussion, I described separately how this model can be addressed in the Backcasting approach "Backcasting approach for everyday application".

For example, Adult patient admitted in the ICU with Ketoacidosis. The patient monitored every 6-8 Hours (sometimes hourly) with arterial blood gas analyze, where included all parameters in our Model. The patient at 8 a.m. was presented with laboratory values: Ph 7.12, Lactate 3.2, HCO3 10, Glucose 417, K 4.9, Na 136, Cl 99, Ca 10.5. Our model shows if untreated a risk of death within 0-8 hours is 99.59%. 

During treatment (fluids, insulin etc.) at 9 a.m. shows with laboratory values: Ph 7.21, Lactate 2.8, HCO3 14, Glucose 325, K 4.2, Na 135, Cl 98, Ca 10.4. calculated a risk of death from 65.72%.

Conclusion: During 1 hour of treatment we have reduced the risk of Patient death from 99.59% to 65.72% based on Deep Learning LEMA-model with AUC of 0.982, specificity of 0.947, and sensitivity of 0.935. 

The R-based interactive web applications for risk adjustment in this link: https://albiondervishi.shinyapps.io/LEMA/

The reviewer’s comment 5. In the Discussion and conclusion, the author should again highlight the importance and shortcomings of the study and be more concrete as to why “deep learning model in recognizing patients at risk for deterioration needs clinical validation”. Are there confounding factors

? Response 5. Thank you for recommendation. I tried to address this issue throw the Manuscript. There are no confounding factors, I supposed it will be a while until the models can be widely used (e.g. APACHE II 1985, SOFA 1996), and still, we perform data analysis based on these models.

The reviewer’s comment 6. Correlation aspects between the features have not been discussed. The author should perform more exploratory analysis of the dataset like PCA, etc.

Response 6. Thank you for this hint. Indeed, it helped me to comprehend better the interaction between parameters. I used the standard Pearson’s correlation matrix. Additionally, I added also the maximum and minimum of my data in Tables 1 and 2.

The reviewer’s comment 7. “Our model uses multilayer neural networks, but a precise evaluation of their architecture would be beyond the scope of this study” - but a precise description of the model architecture/details and chosen parameters and hyper-parameters is a must. Response 7. We thank you for advice. I described the model architecture, chosen parameters and hyper-parameters. Also, I design a Figure (Fig.1) regarding my deep learning model. 

The reviewer’s comment 8. The authors need to compare the deep learning method with some generic baseline model, or some less complex model like Naive Bayes or logistic regression. Response 8. We agree with this and have incorporated your suggestion throughout the manuscript. I compared Deep Learning with a generalized linear model (GLM) as logistic regression and Random Forest. An important conclusion came to a comparison between them.

The reviewer’s comment 9. Why not use something like permutation feature importance? Response 9. We are very grateful to your reflective suggestions. Because I did a correlation together with models’ comparison, I believed it can be enough for parameter interaction and importance to the model.

The reviewer’s comment 10. How does the age and gender of the patients (training set) influence the prediction? Response 10. Thank you for this question. I added in this revision that age and sex were included in the data selection but were not incorporated in the model for risk adjustment. I consider that age and sex are not physiological parameters and I assume that cannot contribute in my model. 

In conclusion, according to the comments, we attempted to revise the original manuscript. During the revised process, I added two Figures (Figures 1 and 4) also, added one table (Table 3). Hope the manuscript has been improved and it has reached the journal standard.

Please feel free to address all correspondence concerning this manuscript to me at albiondervishi@gmail.com

Sincerely,

Albion Dervishi

---

## [Decision Letter · Decision Letter 1]

11 Nov 2020

A Deep Learning Backcasting Approach to the Electrolyte, Metabolite, and Acid-Base Parameters That Predict Risk in ICU Patients

PONE-D-20-09103R1

Dear Dr. Dervishi,

We’re pleased to inform you that your manuscript has been judged scientifically suitable for publication and will be formally accepted for publication once it meets all outstanding technical requirements.

Kind regards,

Feng Luo

Academic Editor

PLOS ONE

Additional Editor Comments (optional):

Reviewers' comments:

Reviewer's Responses to Questions

**Comments to the Author**

1. If the authors have adequately addressed your comments raised in a previous round of review and you feel that this manuscript is now acceptable for publication, you may indicate that here to bypass the “Comments to the Author” section, enter your conflict of interest statement in the “Confidential to Editor” section, and submit your "Accept" recommendation.

Reviewer #1: All comments have been addressed

2. Is the manuscript technically sound, and do the data support the conclusions?

Reviewer #1: Yes

3. Has the statistical analysis been performed appropriately and rigorously? 

Reviewer #1: Yes

4. Have the authors made all data underlying the findings in their manuscript fully available?

Reviewer #1: Yes

5. Is the manuscript presented in an intelligible fashion and written in standard English?

Reviewer #1: Yes

6. Review Comments to the Author

Reviewer #1: The author has indeed addressed all the comments during the first round of the review. The manuscript now appears to be more detailed and rigorous. I would recommend publication. However, I would suggest making the Abstract more concise and less heavy with technical details. The author might look up other papers in Plos one and somewhat adopt the styles.

7. PLOS authors have the option to publish the peer review history of their article (what does this mean?). If published, this will include your full peer review and any attached files.

Reviewer #1: No

---

## [Editor Report · Acceptance letter]

8 Dec 2020

PONE-D-20-09103R1 

A Deep Learning Backcasting Approach to the Electrolyte, Metabolite, and Acid-Base Parameters That Predict Risk in ICU Patients 

Dear Dr. Dervishi:

I'm pleased to inform you that your manuscript has been deemed suitable for publication in PLOS ONE. Congratulations! Your manuscript is now with our production department. 

Kind regards, 

on behalf of

Dr. Feng Luo 

Academic Editor

PLOS ONE